# Perspectives on the Barriers to Nuclear Power Generation in the Philippines: Prospects for Directions in Energy Research in the Global South

Aireen Grace Andal [1,*] , Seepana PraveenKumar [2,*] , Emmanuel Genesis Andal [3], Mohammed A. Qasim [2] and Vladimir Ivanovich Velkin [2]

1    Macquarie School of Social Sciences, Macquarie University, Sydney, NSW 2109, Australia
2    Department of Nuclear and Renewable Energy, Ural Federal University Named after the First President of Russia Boris Yeltsin, 620002 Ekaterinburg, Russia; mkasim@urfu.ru (M.A.Q.); v.i.velkin@urfu.ru (V.I.V.)
3    College of Economics and Management, University of the Philippines Los Banos, Los Baños 4030, Philippines; etandal@up.edu.ph
*    Correspondence: aireengrace.andal@mq.edu.au (A.G.A.); seepanapraveenkumar5@gmail.com (S.P.K.)

**Abstract:** This paper offers a discussion on the social dimensions of the barriers to nuclear power generation in the country. The aim of this paper is to contribute to the literature by identifying the barriers to nuclear power generation in the Philippines and offering perspectives on the social relevance of potentially adding nuclear sources to the country's energy mix. Given the contemporary relevance of the energy transitions globally, this work builds on the available sources over the past decade concerning nuclear energy technology in the Philippines and provides further discussions on the diverse barriers to the country's energy transition pathway. Findings present barriers related to politics, policy, infrastructure, technical capacities, environment and information. The differences in priorities and values concerning nuclear energy reflect that the barriers to nuclear energy generation in the Philippines are social as much as technical. Based on the findings and descriptions of the current discussions on Philippine energy generation, this work provides some key points for consideration in order to deploy nuclear power plants in the country. These recommendations, however, are not definitive measures and are still subject to local conditions that may arise. This study hopes to be instructive to other countries in terms of further reflecting on the social dimensions of the barriers to nuclear energy generation.

**Keywords:** nuclear energy; nuclear power plant; potential barriers; energy generation; energy policy

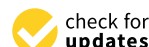



## 1. Introduction

The overall demand for electricity increased globally in the past decade, thereby levying a pressing concern on the direction of energy governance and decisions [1–3]. As such, it has become imperative for countries to transition toward clean and safe energy production, such as generating electricity with technologies from solar, wind, hydro, thermal, and nuclear sources [4,5]. To this end, nuclear power generation remains a highly contested subject as a reliable clean energy source. On the one hand, many countries are generating electricity using nuclear fission technology as it provides energy without releasing greenhouse gases (GHGs). But on the other hand, nuclear plants generate radioactive waste that threatens human life and the environment, which is why such plants are installed in remote areas [6]. Considering such debates, the Republic of the Philippines is one of the countries that seriously weigh the pros and cons of including nuclear sources as part of its energy mix. The Philippines is an archipelagic country in Southeast Asia with about 7,640 islands situated in the western Pacific Ocean. The country has one of the fastest-growing economies, with a growth rate 6.3% per year [7]. In 2020, its Gross National Product (GNP) has been stuck at 362.24 USD [8] due to the COVID-19 pandemic. Forecasts, however, remain positive,

with the country's economy still projected to grow by 5.5% in 2022 [9]. With a growing economy and a population of about 109 million and a growth rate of 1.63% per year [10], the Philippines needs to strategize its energy generation as it will cater to a relatively large population in the coming decades. This population momentum has imposed an additional burden on the country's annual energy consumption and overall energy demand. The country's average energy consumption has been increasing by about 5.5% per annum since 2009 [11], which also holds a record for being one of the fastest-growing consumers of energy in the world as different sectors are expected to increase their energy demand in the next decades (See Table 1) [12]. This projected increase in energy demand has led the Philippine government to venture into modifying its energy mix and including nuclear power as one of the most viable yet controversial sources.

**Table 1.** Energy demand by sector 2020–2040 [12].

| Sector | 2020 (TWh) | 2025 (TWh) | 2030 (TWh) | 2035 (TWh) | 2040 (TWh) |
|---|---|---|---|---|---|
| Residential | 27.8 | 34.3 | 42.4 | 52.2 | 64.4 |
| Commercial | 24.8 | 30.7 | 37.9 | 46.7 | 57.6 |
| Industry | 28.4 | 35.2 | 43.3 | 53.4 | 65.7 |
| Others | 2.9 | 3.6 | 4.4 | 5.4 | 6.7 |
| **Total** | 83.9 | 103.8 | 128 | 157.7 | 194.4 |

*1.1. Global Context*

The world has witnessed the potential of the Philippines to be a leader of nuclear energy generation in Southeast Asia, as it has the first nuclear power plant in the region. The country started to build a nuclear power plant in 1976 but after the completion of the 623-megawatt (MW) Bataan Nuclear Power Plant (BNPP) in 1984, the mission to operate it was aborted [13]. Since the BNPP has been mothballed, the Philippines has not attempted to re-open it or build other nuclear power plants in the country, leaving it behind other countries that generate nuclear power. There are currently 444 nuclear reactors in operation worldwide, and 51 more are under construction. Table 2 presents the list of countries with the largest number of nuclear power plants in the world [14–16], with no representation from Southeast Asia. As such, the Philippines still has to make a decision on its national position on nuclear energy. Drawing mainly from fossil fuels, geothermal, wind, hydropower and solar energy [17,18], it might be difficult for the country to meet the needs of the citizens to have a lower cost of electricity. Since the establishment of the BNPP and its immediate withholding, nuclear energy generation in the Philippines has been controversial. Aside from the $1.2 billion foreign debt the Philippines still owes for BNPP, holding off its use was imbued with the political and risk issues of Chernobyl at the time when Corazon Aquino took the presidency after Ferdinand Marcos. There were attempts to raise discussions on re-opening the nuclear power plant, especially in the 1990s when the Philippines was on the brink of a power crisis and in the 2000s when there was an enormous increase in oil prices. To date, the Philippine government still pays up to $1 million a year to maintain the power plant in "preservation mode" [19]. The country does not have any new nuclear power plants and the issue of whether to pursue nuclear energy generation is still highly debated. The IAEA advises that the former NPP be given a thorough evaluation both in technical and economic terms [14].

**Table 2.** Status of Nuclear power reactors. Data source: Asian Development Bank.

| Country | Number of Operable Nuclear Reactors | Under Construction Nuclear Reactors | Total Net Electrical Capacity of Plants under Construction (MW) |
|---|---|---|---|
| United States | 93 | 2 | 2234 |
| France | 56 | 1 | 1630 |
| China | 51 | 14 | 13,765 |
| Russia | 38 | 3 | 3459 |
| Japan | 33 | 2 | 2653 |
| South Korea | 24 | 4 | 5360 |
| India | 23 | 6 | 4194 |
| Canada | 19 | 0 | - |
| United Kingdom | 15 | 2 | 3260 |
| Ukraine | 15 | 2 | 2070 |
| Belgium | 7 | 0 | - |
| Spain | 7 | 0 | - |
| Czechia | 6 | 0 | - |

*1.2. Current Local Context*

In recent years, there was a renewed interest and developments in nuclear energy in the Philippines. In July 2020, President Rodrigo Duterte (30 June 2016 to 29 June 2022 in office) signed the Executive Order (EO) 116, an issuance to develop the Civilian Nuclear Energy (CNE) and Nuclear Energy Program Inter-Agency Committee (NEP-IAC) to lead the Nuclear Power Program (NPP) for the Philippines, including viability studies of nuclear energy in the country, and further recommendations toward introducing nuclear energy in the country's energy landscape [20]. The Department of Energy (DoE) of the Philippines has also signed a Memorandum of Intent with Rosatom Overseas JSC on having pre-feasibility research of nuclear power plants based on small modular reactor technology [21]. Such plans are in line with the goals of the Philippine Energy Plan (PEP) which will include nuclear sources in the energy mix by 2040, which is approximately comprised of a 70% base generation capacity from coal, geothermal, hydropower, natural gas, nuclear and biomass, 20% mid-merit capacity from natural gas, and 10% capacity for peaking from oil-based plants and VRE, such as solar PV and wind sources [7,22]. In addition, this case study evolved potential barriers and policy implications and a few recommendations were suggested to develop the nuclear power plant in the Philippines.

*1.3. Overview of the Current Power Generation Schemes in the Philippines*

A perpetual social concern in the Philippines is access to energy. There are around 11 million people without proper electricity facilities in the Philippines and those with access experience frequent electricity fluctuations. Further to this, the Philippines has one of the most expensive electricity prices in Asia, reaching an average retail rate of US $0.15 per kWh [23]. As part of addressing this perpetual social concern, the Philippine government has implemented various programs and policies committed to improving the access for Filipinos to affordable energy, along with reducing carbon emissions and addressing energy security in the country. For instance, consumer-oriented mechanisms are stipulated under the Republic Act No. 9513 (RA 9513), also known as, "An Act Promoting the Development, Utilization and Commercialization of Renewable Energy Resources and for Other Purposes" [24], institutionalised energy programs not only to increase the efficiency of power generation throughout the country but also to alleviate the burden on Filipinos in shouldering the price of electricity. One of the consumer-oriented incentive schemes of RA 9513 is the net metering program, in which an end-user can use the electric power generated from a valid on-site renewable energy facility in order to offset the electric energy provided by direct utility (DU) to the end-user. This allows energy consumers to generate their electricity from renewable energy sources without needing to feed electricity back to the local distribution grid, thereby exempting them from the net electricity bill [25].

Since its pilot with one end-user in 2013, the net metering program is rapidly growing and is increasingly attracting customers in the country [26]. As of December 2020, the net metering program has 3795 registered participants and it is expected to grow even more yearly [27]. Another consumer-oriented scheme is the Green Energy Option, which allows households to choose RE resources as their source for energy consumption. The DoE has invested in training programs, workshops, and other capacity building activities for the local communities, motivating them to choose RE resources [28]. It is also to be noted that the Philippines is a contributor to the drafting and review of IAEA's "Building a National Position for a New Nuclear Power Programme". Being a participant in creating this document is indicative of the importance of the national position for the Philippines, at least during the time of former President Duterte [29]. The type of technology to be used still remains to be seen given that the discussions on nuclear power plants in the Philippines have been superseded by the 2022 national elections, leading to a shift in leadership and priorities. However, newly elected President Ferdiannd Marcos Jr. hints at bringing nuclear power to the Philippines [30].

Whilst the DoE intends to expand the programs of RA 9513 by developing all options for energy sources including nuclear energy, the Philippines faces considerable barriers to the inclusion of nuclear energy. Drawing from the previously mentioned endeavours of the Philippines with regard to nuclear energy, this paper offers a discussion on the social dimensions of the barriers to nuclear power generation in the country. The aim of this paper is to contribute to the literature by identifying the barriers to nuclear power generation in the Philippines and offering perspectives on the social relevance of potentially adding nuclear sources to the country's energy mix. Based on the findings and descriptions of the current discussions on Philippine energy generation, this work then discusses key areas for consideration and social implications with regard to the transition pathways toward adding nuclear sources to the Philippine energy system. The discussions are organised as follows: the next section presents the methodology of this work, followed by the results of the analysis, and then the discussion section offers the recommendations and implications concerning the barriers to nuclear energy generation in the Philippines. The last section concludes by outlining key areas for further attention and examination.

## 2. Materials and Methods

To identify the socially-relevant issues on nuclear power generation in the Philippines, this article conducts an exploratory document analysis from the literature, government sources and mass media to examine the barriers to nuclear power generation in the Philippines. Given the contemporary relevance of the energy transitions globally, this work builds on the available sources over the past decade concerning nuclear energy technology in the Philippines and provides further discussions on the diverse barriers to the country's energy transition pathway. The materials included in this study were identified through a cursory search of keywords, using search engines, government websites, mass media sources, and academic journal publishers accessed via the institutional subscription of the authors' university. Furthermore, the present study searched for key informant interviews available for public access, as well as established news agencies to identify recent events and non-academic reports on issues concerning nuclear energy introduction in the Philippines. All selected materials were in English, including books, academic articles, news reports, and documentaries with content related to nuclear energy discussions in the Philippines. The period studied in this work covers every decade from 2010 to 2020. After the materials were screened and checked for eligibility, the present study examined the content using thematic analysis. After the materials were screened and checked for eligibility, the present study examined the content using the six barriers: political, policy, infrastructure, technical, environmental and information.

### 3. Results: Main Barriers in the Generation of Nuclear Power in the Philippines

This section discusses the various barriers and challenges that curtail the deployment of nuclear power plants in the Philippines. The above-mentioned studies note major challenges, such as (1) high initial cost investment, (2) uncertainty in politics, (3) uncertainty in policy, (4) lack of manufacturing units, and (5) lack of local community awareness. Building on such literature, the following subsections offer the current discussions on the socially relevant context of nuclear energy generation in the Philippines and provide some key points for consideration in order to deploy nuclear power plants in the country. In surveying the literature, the most common issues that the Philippines face concerning nuclear energy generation are a combination of issues, which implies that there is no single issue that stands out. Figure 1 summarizes the key barriers to adopting nuclear power in the Philippines. However, the risk map is highlighted in the Appendix A.

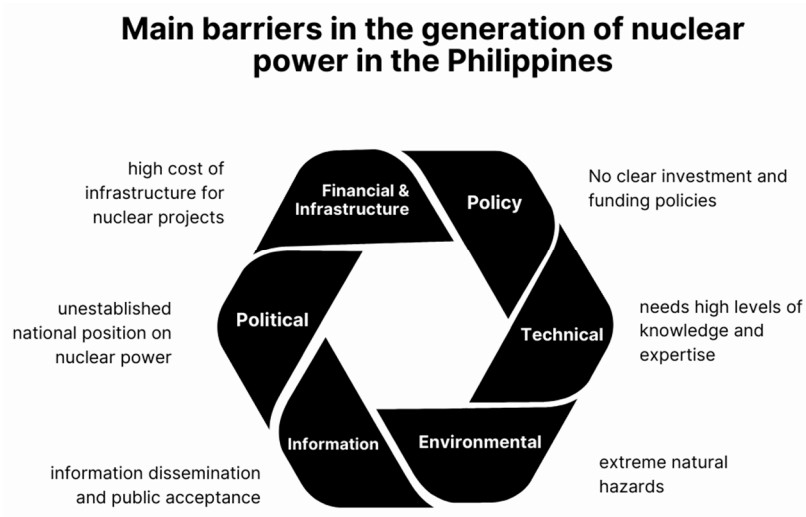

**Figure 1.** Key barriers to adopting nuclear power in the Philippines.

#### 3.1. Political Barriers

In legal developments, the House Bill (HB) 8733, also known as the "Comprehensive Nuclear Regulation Act", aimed at establishing a comprehensive nuclear regulatory framework and creating the Philippine Nuclear Regulatory Commission (PNRC). The bill has passed the second reading in Congress in 2019 and has been referred to the House of Representatives for concurrence [31]. This was one of the many nuclear-related legal steps that has been signed after the Republic Act No. 5207 AKA Atomic Energy Regulatory and Liability Act of 1968 and Presidential Decree No. 1484. However, the major political barrier in the Philippines is its lack of national position on nuclear power generation. DoE Undersecretary Felix William Fuentebella (Fuentabella is an undersecretary of DoE at the time of writing this article) communicated that before making any decisions to pursue nuclear power endeavours, the Philippines must have a firm national position on nuclear energy use [32] having an unclear national position on nuclear power has a long history related to unsettled political tensions between national authorities who support or oppose nuclear energy. The discussions concerning nuclear energy in the Philippines have historically been, and continue to be, influenced by various economic and political powers [33]. Such a structural and bureaucratic lack of direction leaves the Philippines in a stand-off position when it comes to nuclear energy.

This lack of national position can be attributed to diverging opinions from various groups, which have increasingly participated in political and public forums about environmental risks, including nuclear risks. What has drawn together the community of concerned citizens was to push the current administration not to pursue nuclear power generation [34]. Greenpeace Philippines is particularly vocal about focusing on renewable energy [35]. In this group, there is a small sub-group of academics who are generally

divided on the nuclear energy issue. One side shares their suggestions on finding other means toward energy transition on top of their hostility to an extractive economy. To them, nuclear-induced risks to the environment and health go together with the operation of industrial capitalism ab initio. As such, they have been active in the public dissemination of this perspective. With their voice and mobility, the environment advocates created a distinct image and have gained supporters. Because nuclear energy is a controversial topic not only in the Philippines but globally, such groups could easily gain the attention of the popular media due to their established image and communication skills, as well as because they have gained sympathy from leading figures in the Philippines. Furthermore, aside from political figures and environmental groups, religious authorities also play part in the conversations. Safety issues were also raised by the Catholic Bishops Conference of the Philippines (CBCP) [36], which is highly influential in the country given its Roman Catholic majority. Finally, social media plays a role in terms of information dissemination, whereby policymakers can make use of such platforms to spread awareness on the protection against risks towards further acceptance from the public [37]. This is especially pertinent to the younger generations who are usually active users of social media.

### 3.2. Policy Barriers

The country's DoE is explicit in its desire to bring nuclear power into the energy mix of the country. The DoE's Nuclear Energy Programme Implementing Organization (DoE-NEPIO) has hosted several events to publicise the feasibility of nuclear power generation. The agency has even discussed possibilities of joint endeavours with private companies, nuclear energy advocates, such as Michael Shellenberger [38], and other neighbouring countries in the ASEAN to work toward securing a sustainable energy future by adopting nuclear energy [39]. This is in-line with the literature on the role of nuclear energy in the reduction of dependency on greenhouse gas emissions and import energy but this has not taken shape yet.

Major policy barriers include public investments and market facilitation policies. The high cost of nuclear energy results in difficult cost-driven decisions, which stall policy-making. While the current stance of the DoE is clear that nuclear energy will be beneficial in the long term, the opposing views are strong enough to make arguments and delay decisions, such as setting specific public funds for direct investments or for guarantees and capacity-building to facilitate investments.

In terms of investment and funding policies, there is no clear direction for the profitability of nuclear energy generation in the Philippines, making it difficult for authorities to finalise a decision. Mendoza et al. [40] provided an account of the BNPP as a large-scale but unprofitable investment project that burdens businesses, the government and citizens. Meanwhile, Agaton [41] conducted a Real Options Approach (ROA) to analyse investment values and optimal timing for the technology transition from coal to nuclear energy and renewable sources, which provided evidence of welfare losses from delayed investment in alternative energy sources. The DoE also carried out a sensitivity analysis showing that the scheduled onset of nuclear power generation depends on the level of electricity demand and investment cost. Moreover, sentiments raise that the introduction of nuclear energy will be "a great burden" [42] for Filipinos. While the long-term benefits of adding nuclear energy to the Philippine energy mix are promising, the question is whether this is the optimal option for the Philippines. The high cost of such an enterprise gives very little incentive to the private sector to invest in nuclear energy. However, if the government chooses to subsidise this entire endeavour, taxpayers might be divided to support such a costly investment. Paying an additional ten centavos per kilowatt-hour for at least five years may burden poor taxpayers who would have to pay such a monthly nuclear tax [33].

### 3.3. Financial and Infrastructure Barriers

While studies have already proven that nuclear energy is more economical for developing countries that import energy resources, the archipelagic nature of the Philippines

creates some transportation challenges for operating nuclear power plants. As of the moment, there is no concrete plan for market facilitation that encourages and supports market participants to engage in nuclear energy technology deployment.

Moreover, the cost of infrastructure for nuclear projects might be a barrier in the Philippine market in which private investors expect a short-term return on investment or immediate paybacks. This is especially relevant as the Philippines is still under a period of economic recovery from the impact of the COVID-19 pandemic. Such large-scale and long-term nuclear energy infrastructure projects may not be a priority for both the government and private sectors.

Additionally, the country needs to develop quality infrastructure to manage nuclear waste. Whereas nuclear power is efficient, clean, and has high energy density, it shows a promising trajectory in the long term, the issues faced by the Philippines are related to nuclear waste management and high start-up/investment costs, which does not attract the private sector to invest. The PNRI has a Radioactive Waste Disposal Program with activities, such as a National Radwaste Repository Centre, conditioning of disused radioactive sources, treatment and storage of radioactive waste, and co-location of near surface facility studies [43]. However, there is still a need to have further training and practice to operate a nuclear power plant in the Philippines. As the Senate Economic Planning Office (SETO) of the Philippines stated in its policy brief, "To operate an NPP without building up the technical and regulatory capabilities needed to ensure its safe and efficient operation is not only unwise but extremely dangerous and risky as well" [44].

*3.4. Technical Barriers*

Nuclear power generation in a country needs high levels of knowledge and expertise. While Filipino scientists have developed competitive expertise in renewable energy sources, the Philippines need more experts with the ability to manage important domains in the organisation of a nuclear power plant. As of the moment, only the Philippine Nuclear Research Institute (PNRI) conducts most activities on research and development in the field of nuclear sciences and technology [45]. There are no private institutions that implement research in this field. Yet there is a lot to learn in terms of the viability of nuclear power generation in the Philippines considering its safety, major reactor accidents, disposal of nuclear waste, and even the risk of nuclear weapons proliferation [46]. Moreover, there is scarce literature on operating nuclear power in the Philippines with respect to energy security, price volatility, response to climate change issues, updates in public perception and acceptance as well as knowledge on putting up nuclear medicine centres [47].

Filipino scientists have not developed the technical competence to independently operate nuclear power. In terms of keeping relevant technical discussions with the wider global community, the Philippines remains participative in international discussions on matters of the energy mix and sustainable development. Aside from being a very active member of the International Atomic Energy Agency for the past 20 years, the Philippines also works closely with various partners concerning broad energy-related activities and international events about climate change. The country is also open to collaboration in trying various pathways toward transitioning away from fossil fuels as energy sources. Like other Southeast Asian countries, the Philippines also actively presents national reports about its national context and position on global environmental problems and initiatives regarding nuclear technologies. However, the country does not have the technical capacity to perform its own studies independently. Marquardt et al. [48] have ventured into examining foreign aid in the Philippines and Morocco to discuss the opportunities and barriers of supporting energy transitions, including nuclear energy, which suggests that such countries might struggle without financial and technical aid. In addition, while international cooperation on capacity-building with other countries is ongoing, there is still no clear direction on the extent of such partnerships. For instance, Yap [49] mentions that one option for the Philippines is to invest in small modular reactors (SMRs). While SMRs will provide stable electricity supplies to the off-grid or remote island communities [50] as it does not require

large-scale grid infrastructure [51], Filipino scientists still need extensive technical guidance to operate such technologies.

### 3.5. Environmental Barriers

The main environmental concern for the Philippines in installing nuclear power plants is location. In the case of the existing Bataan Nuclear Power Plant, studies have long-standing debates concerning the risk of operating it because of its proximity to Mt. Natib, exposing it to volcanic hazards. Whereas Venida and Reyes [52] provided justifications for the safety of BNPP's location, Lagmay et al. [53] have also published evidence that the BNPP's location is unsafe as it is exposed to geological hazards. Yet in an interview for their systematic study using electrical resistivity, seismic refraction and radon gas detection, Arcilla et al. [54] claimed that there are no active faults that lie beneath the BNPP.

Environmental externalities play a big role in opening other nuclear power plants elsewhere in the country. Extreme natural hazards in the Philippines can initiate nuclear accidents. Being situated in the Pacific, the Philippines is always exposed to tropical cyclones (TCs), with an average of 20 TCs entering the Philippine Area of Responsibility (PAR) per year [55]. Floods can damage nuclear sites and severely impair the necessary systems needed to support and sustain safety functions and provide backup power or generators. Seawater can also overload the reactor buildings with a large amount of debris, restricting access to the site just as what happened to the Fukushima Daiichi nuclear power plant accident in March 2011 [56]. This has sparked many academic discussions and public debates about whether nuclear energy is optimal for the Philippines [57].

Moreover, many sites in the Philippines are also prone to earthquakes from active seismotectonic conditions as it is located on the Pacific Ring of Fire. There are at least 65 active or potentially active volcanoes surrounding the Philippine Islands and there were over 90 earthquake events with a moment magnitude (Mw) >7.0 recorded since the 1960s [58]. Such earthquake conditions can damage electricity transmission lines and substations in nuclear power plants, which will inevitably result in a total loss of power, similar to the Fukushima accident. In the latter's case, both an earthquake and tsunami caused loss of electricity in Unit 1, 2, and 4 reactors and a blackout in Unit 3, succeeded by a massive radioactive material discharge from Unit 2 due to damage suppression chamber as well as a hydrogen explosion at the reactor building of Unit 4 [59]. Due to the risk of disasters, it is considerably challenging for the Philippines to guarantee a safe site, along with developing essential safety systems that will prevent damage to the nuclear reactors and the discharge of radioactivity into the environment.

### 3.6. Information Barriers

Decisions on nuclear energy in the Philippines have remained largely technical, discussed by scientists and technicians, although decisions and policies on issues of nuclear safety are publicised through government documents. The majority of the technical discussions on nuclear energy in the Philippines are led by its Department of Energy. In terms of policies, the leading opinions concerning nuclear energy include two national regulatory authorities in the Philippines responsible for nuclear energy use: (1) the Philippine Nuclear Research Institute (PNRI) under the Department of Science and Technology (DOST), which is in charge of the regulation, licensing and safeguards of radioactive materials and atomic energy facilities and (2) the Bureau of Health Devices and Technology (BHDT) under the Department of Health (DOH), which is responsible for radiation protection and safety of the ionising and non-ionizing radiation emitted from electrical/electronic devices [60].

Apparently, public acceptance is not the biggest issue when it comes to the pursuit of nuclear energy in the Philippines. However, a caveat needs to be addressed. While the majority of Filipinos (79%) support investing in the Nuclear Power Program, DoE Secretary Alfonso Cusi expressed, "The only problem is that [they] do not want to have a nuclear power plant in [their] own backyard" [26]. This goes in line with the initial justifications on the non-operation of the first nuclear power plant in Bataan. Concern

about risks of location is a major driver of discussions on nuclear energy development in the Philippines. Further to this, Ong et al. [61] just released their study on the acceptance of the reopening Bataan Nuclear Power Plant, which reports that risk-averse individuals tend to negatively view the BNPP reopening and those who focus on the benefits have positive acceptance. Such risk aversion can be related to the meltdown at Chernobyl and the tragedy of Fukushima, which continue to serve as a precaution to the point where, in the Philippines, nuclear development has not been a viable energy option. Regardless of studies on the probability of risk, the view that nuclear plants are unsafe remains dominant in the discussions of energy transitions in the Philippines. The fear of risks of nuclear power plants is prevalent among leadership groups in the Philippines, including politicians, government bureaucrats and diplomats, national-media journalists, public interest groups as well as several scientists. There is a dearth of studies conducted on Filipinos' acceptance of nuclear energy in their country. Nevertheless, as the study of Ong et al. [61] indicates, the level of acceptance in terms of perception of benefit versus risk, Filipinos tend to gravitate toward the perceived benefits.

## 4. Discussion: Key Areas for Consideration for Social Aspects in Energy Research

From the standpoint of international environmental justice, the energy demand of the Philippines is critical as it has one of the largest populations in the world, yet many citizens face the precariousness of paying a high price for energy. From a local standpoint, the Philippines has a unique combination of barriers to nuclear energy generation from a technical level to political conditions to geographical characteristics. As energy demand increases, there needs to be a balance between a feasible and cheap energy source and also a sustainable and locally-appropriate energy source. Energy technologies have varied widely in their commercial status and also technological maturity, and the challenge is to decide which energy technologies are best to cater to the energy needs of a growing population. As such, below are some targeted recommendations to address the above-mentioned barriers and challenges. These recommendations, however, are not definitive measures and are still subject to local conditions that may arise.

### 4.1. A National Position

As previously stated, the DoE calls for clarifying the national position on nuclear power generation in the Philippines. As such, there is a need to have further consultation meetings and discussions not only among experts but with different stakeholders, especially those who are in opposition to nuclear power generation in the Philippines. As recommended by Masinas et al. [62], a transparent relationship and two-way communications channel are crucial for awareness. This extends to those who are against establishing nuclear power plants in the Philippines. This is especially pertinent to the agenda of the Philippines to reduce its net import of fossil fuels for sustainable development goals [63]. The proponents of nuclear energy need to show the alignment of nuclear energy to the goals of replacing burning fossil fuels for energy in order to reduce carbon dioxide emissions, thereby mitigating the impact of climate change, such as risks of disasters and extreme weather conditions [64].

### 4.2. Pro-Poor Policies

Since Executive Order No. 116 (EO 116) has already been signed by President Duterte, it is in the best interest of the DoE to expedite studying the adoption of a national nuclear energy policy. This policy agenda is already within the Nuclear Power Program Roadmap of DoE-NEPIO. It is then necessary that to indeed achieve lower costs, nuclear reactors should be standardised, and that an international value chain be developed to take advantage of lower labour costs in other countries. The full gains from cost reduction, however, can only be taken advantage of if the public has enough confidence in nuclear power to allow the recalibration of the current planning and regulatory system [65]. As has already been initiated by DoE, this is to be followed by immediate action from an independent

regulatory body to guarantee the safety of nuclear power through the ratification of related treaties and conventions, as well as the establishment of bilateral agreements with international suppliers, and develop policies, guidelines and procedures for procurement and contracting procedures.

### 4.3. International Cooperation

Taking into consideration that the development of infrastructures for nuclear energy can be challenging for starting countries, this requires a major effort for the Philippines to cooperate with countries whose experience in nuclear energy generation is already advanced. Moreover, it makes sense for the country to collaborate regionally with other Southeast Asian countries so that they can share the cost of developing the necessary infrastructure. This international and regional cooperation could be even more beneficial for energy supply operations from the public sector to private sector companies. Furthermore, the Philippines can also invest more in large-scale hydroelectric power to combine with nuclear energy "which can supply a vast amount of stable and carbon dioxide-free power" [38].

The DoE established further collaborations with private sectors to conduct studies regarding nuclear power generation. For instance, in 2019, DoE partnered with Korea Hydro & Nuclear Power Co., Ltd. (Gyeongju, Korea) in a Pre-Feasibility Study for the deployment of SMR in the Cagayan Economic Zone Authority (CEZA), (KHNP) [61]. Likewise, the DoE and Rusatom Overseas JSC signed a Memorandum of Intent on cooperation regarding small modular reactors, and in 2021, Rusatom hosted Filipino scientists on a virtual tour at Leningradskaya nuclear power plant [66]. Prior to such collaboration is the engagement of the Philippines in the Integrated Nuclear Infrastructure Review (INIR) mission in 2018, in which experts from the IAEA reviewed the status of the Philippine nuclear infrastructure using the criteria for the first phase of the IAEA's Milestones Approach [67].

Regarding knowledge and training on waste management, efficient burying of waste is not unique to nuclear waste and is found in any current technology. Filipino scientists can improve their knowledge through research, which is already in place both in terms of technology development and policy strategies [68]. At the moment, there is a growing discussion on the ethical dimensions of nuclear waste management policies [69] Moreover, studies are also exploring thorium power plants, which have more manageable wastes because thorium degenerates into more stable isotopes, unlike uranium and plutonium [70,71]. Moreover, those were just with regards to nuclear fission technology. When research in nuclear fusion develops further, a more optimistic pathway opens up for the Philippines to turn to nuclear energy. Fusion can be a solution in the long run, and thorium plants in the short run. Along with such developments in nuclear research, there is unquestionable progress in other renewable sources and huge innovations are on the way as energy keeps on getting cheaper and cheaper as well. Another prospective solution is for the Philippine government to collaborate with the private sector, both foreign and local [72]. The point is that adding nuclear to the Philippine energy mix provides a diversity of options as a way to transition away from fossil fuels.

### 4.4. Risks

One of the strongest oppositions to establishing nuclear power plants in the Philippines is the fear of environmental risks as informed by history. However, risks are not solely found in nuclear power plants and data show that nuclear energy produced the least casualties per kilowatt hour of energy generated [73]. In terms of environmental consequences, nuclear power plants are unlikely to be a major exacerbator of climate change problems [74]. Besides, the anxiety over nuclear accidents and meltdowns can be considered as lessons from the past. In taking a closer look at the literature, the positive externalities of decarbonization from nuclear power are on par with those of renewable energy [75]. One way to address potential risks is to assess the country's safety culture, specifically the difference between the perception of the benefits and risks of nuclear energy [76]. Another way is to conduct the TRL-technology readiness level [77], which can be used as a reference for assessing risks

and decision making. This also makes a case that the fear of nuclear power plants being used for the proliferation of nuclear weapons needs further solid ground to stand on. The high cost of nuclear power investment, and the limited isotopes for energy consumption which are difficult to enrich, do not incentivize producing weapons. While we cannot discount the possibility of using nuclear power for weapons, no current data support this probability. In fact, a study conducted by Brutschin et al. [78] revealed that the pursuit of nuclear weapons only has a minor role in a country's decision to pursue nuclear technology.

*4.5. Selection of Potential Site*

Previous research suggests that the BNPP has general hazard concerns, such as its proximity to the potentially active volcano, Mt. Natib, which is aligned with the Lubao Fault whose traces have high radon emissions [53]. Alternatively, the sub-region of Cagayan in Northern Luzon is deemed a promising site for an NPP in the Philippines. This location has been mentioned in various studies as a potential site for establishing a nuclear power plant. In the 1990s, the National Power Corporation (NaPoCor) led a site investigation on a detailed evaluation of the province of Cagayan as a site for nuclear power generation [79]. The Siting TWG of the DoE-NEPIO has even already conducted a field verification survey and ocular inspection of the proposed sites in Sta. Ana, Cagayan [80]. Furthermore, the Korean company, Korea Hydro & Nuclear Power (KHNP) (Gyeongju, Korea), is considering investing in a 100-megawatt modular nuclear reactor at Cagayan Economic Zone Authority (CEZA) in Sta. Ana, Cagayan [49]. Likewise, the PNRI has conducted a study on the Preferred Site for Co-location of Near Surface and Borehole Disposal of Spent Sources (BOSS) Facility, to which the proposed disposal site is located in Northern Luzon with 40 hectares for potential development [44].

In terms of geological hazards, Cagayan has a very low risk of temperature increase but has a medium risk in terms of projected rainfall. The climate of Sta. Ana has an even distribution of rainfall throughout the year. The months between March and late July encompass the dry season, with the highest temperature at 35 degrees Celsius. Occasional rains occur during the months of June, July and August. As such, some issues with putting up an NPP are related to flooding as the northeast monsoon weather produces rainfall that has an annual average of 26 mm. Overall, Cagayan was assessed to have a low risk for geophysical disasters [81]. Besides, the province has been proactive in Strengthening of Disaster Management Capability/System, especially in flood monitoring, forecasting and warning systems [82]. Furthermore, transportation of items will be secured as this location is a major transshipment point in the Asia-Pacific, located between the Pacific Ocean and the South China Sea. The area also guarantees a consistent power supply and distribution system [83].

Aside from Cagayan, there are other sites that The Nuclear Energy Program Inter-Agency Committee (NEP-IAC) considers, such as Concepcion and Tagbarungis in Puerto Princesa City; Picaon Point and Cauit in Siocon, Zamboanga Del Norte; San Carlos City and Sipalay in Negros Occidental and Bayawan City, Negros Oriental. Moreover, much earlier studies of NEP-IAC between the 1970s and 1995 noted some sites, such as Bagac, Bataan; San Juan, Batangas; Padre Burgos, Quezon; Ternate, Cavite and General Santos City. However, their suitability as potential sites are still under review until now [84]. As of August 2020, the DoE head talked about the potential of introducing nuclear energy in Cagayan, Palawan, and Sulu [84,85].

## 5. Conclusions

The aim of this article has been to advance and contribute to the understanding of the energy transition in the Philippines. The findings in this study suggest that the differences in priorities and values concerning nuclear energy reflect that the barriers to nuclear energy generation in the Philippines are social in addition to technical. This calls for further drawing together diverse perspectives and discussing important points for the governance of energy transitions in the Philippines. This also requires considering the many layers

involved in the process of decarbonization in the country, such as the consequences for marginalized groups [86]. The materials examined in this study provide a glimpse of the intrinsic links among political, economic, social and environmental aspects of pursuing nuclear power in the Philippines. Decisions, actions and consequences in any one aspect will have an impact on the other dimensions. A dwindling natural environment presents challenges to survival and is equally unlikely to sustain a falling economy [75].

The next question is thus where the contemporary debates might lead Filipinos with regards to the pursuit of nuclear energy, for which there is no easy answer. Indeed, the introduction of nuclear energy in the Philippines can be a game changer for many lives, which needs a holistic approach toward the ultimate improvement of Filipinos' quality of living. It is thus important to draw on both social science and natural science research in responding to technical, structural, institutional, and socio-political dimensions of nuclear energy generation in the Philippines [87]. The diverging views on the introduction and development of nuclear energy have emerged from a specific local context related not only to economic and environmental concerns, but also to political ideologies. Currently, efforts of nuclear energy generation in the Philippines will take shape if followed by studies that identify more sophisticated ways of responding to polarized views on nuclear energy introduction in the Philippines and its expected impact. As such, if more sustainable energy supply network designs are to be adopted by the Philippines, there needs to be harmony among all dimensions for efficient and fast implementation of future energy projects. Such responsibility would make relevant and timely contributions to the broader research drive on establishing interdisciplinary research synergies. These observations deserve careful judgment, discussion, and decisions. This work thus invites further transdisciplinary studies on other countries to be compared to the Philippine context.

**Author Contributions:** Conceptualization, A.G.A., S.P.K. and E.G.A.; methodology, A.G.A.; software, A.G.A. and S.P.K.; validation, A.G.A., E.G.A. and V.I.V.; formal analysis, A.G.A.; investigation, A.G.A., S.P.K.; resources, A.G.A., S.P.K., E.G.A., M.A.Q.; data curation, A.G.A., S.P.K., E.G.A.; writing—A.G.A., S.P.K., E.G.A., writing—review and editing, A.G.A. and E.G.A.; visualization, A.G.A., S.P.K.; supervision, E.G.A. and V.I.V.; project administration, M.A.Q.; funding acquisition, not applicable. All authors have read and agreed to the published version of the manuscript.

**Funding:** This research received no specific grant from any funding agency in the public, commercial, or not-for-profit sectors.

**Acknowledgments:** Authors would like to express sincere thanks to the anonymous reviewers for sharing their expertise that greatly assisted the revision of this work.

**Conflicts of Interest:** The authors declare no conflict of interest.

## Appendix A

Figure A1 shows the risk environment map concerning the installation of nuclear power plants in the Philippines.

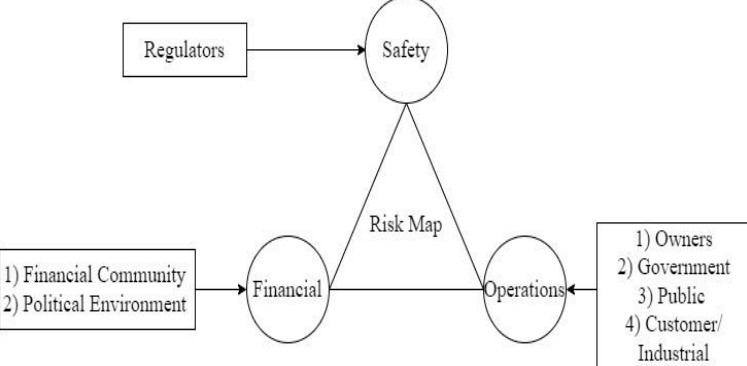

**Figure A1.** Risk management environment model for a nuclear power plant operator.

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
