# Peer review of "Perspectives on the Barriers to Nuclear Power Generation in the Philippines: Prospects for Directions in Energy Research in the Global South"

_inventions, doi:10.3390/inventions7030053_

Round 1
Reviewer 1 Report
1) Commend the authors for relevant paper as submitted. It is mostly well written but this reviewer suggests additional consideration of the following:
1a) risk map and risk perception works by Paul Slovic; separately by Tversky and Kahneman on uncertainties, also risk and society by C. Starr. What is the risk map for the Philippines? Inclusion of a risk map may be appropriate.
2) It has a good set of references.
3) Also, what is the estimated level of acceptance in terms of the perception of benefit versus the perception of risk, in the Philippines. One sentence for some of the references, [73, 74, 75] does not seem to sufficient.
4) What is the role of social media in developing public acceptance, social license and/or public opinion? Granted, these are difficult topics but unless deeper consideration is given, we do not make progress. This is the opinion of this reviewer.
Author Response
Authors will want to express our profound gratitude to you and the respected Reviewers for the swift and in-depth nature of the reviewing process. The contributions and queries from reviewers have helped improve this paper to achieve its objective. Enclosed with this letter you will find an electronic submission of a revised manuscript titled “ Perspectives on the social dimensions of barriers to nuclear power generation in the Philippines: Prospects for directions in energy research in the Global South ” by Aireen Andal Grace, Seepana PraveenKumar, Emmanuel Genesis Andal, Mohammed

Reviewer 2 Report
The paper provides sufficient information on the status of nuclear activities and perspectives in the Philippines. What is missing though are technical details regarding the new nuclear power plants. To begin with, it would be desirable to state what the power of such a power plant would be, because we see that the Philippines is expecting a significant increase in demand for electricity in the next 20 years.
Something could also be written about the technology, ie. what type the new nuclear power plant would be. The paper mentions Korean and Russian technologies and that the focus is on SMRs. These are new, low-power, technologies. Would such a power plant have sufficient capacity for Philippine needs?
Author Response

(The authors gave the same response as above.)
